# Combinatorial microRNA Loading into Extracellular Vesicles for Increased Anti-Inflammatory Efficacy

**DOI:** 10.3390/ncrna8050071

**Published:** 2022-10-21

**Authors:** Alex Eli Pottash, Daniel Levy, Anjana Jeyaram, Leo Kuo, Stephanie M. Kronstadt, Wei Chao, Steven M. Jay

**Affiliations:** 1Fischell Department of Bioengineering, University of Maryland, 8278 Paint Branch Drive, College Park, MD 20742, USA; 2Translational Research Program, Department of Anesthesiology and Center for Shock, Trauma and Anesthesiology Research, University of Maryland School of Medicine, 660 West Redwood Street, Baltimore, MD 21201, USA; 3Program in Molecular and Cell Biology, University of Maryland, 4062 Campus Drive, College Park, MD 20742, USA

**Keywords:** exosomes, miRNA, inflammation, sepsis

## Abstract

Extracellular vesicles (EVs) have emerged as promising therapeutic entities in part due to their potential to regulate multiple signaling pathways in target cells. This potential is derived from the broad array of constituent and/or cargo molecules associated with EVs. Among these, microRNAs (miRNAs) are commonly implicated as important and have been associated with a wide variety of EV-induced biological phenomena. While controlled loading of single miRNAs is a well-documented approach for enhancing EV bioactivity, loading of multiple miRNAs has not been fully leveraged to maximize the potential of EV-based therapies. Here, an established approach to extrinsic nucleic acid loading of EVs, sonication, was utilized to load multiple miRNAs in HEK293T EVs. Combinations of miRNAs were compared to single miRNAs with respect to anti-inflammatory outcomes in assays of increasing stringency, with the combination of miR-146a, miR-155, and miR-223 found to have the most potential amongst the tested groups.

## 1. Introduction

Inflammation-related diseases are responsible for millions of deaths every year [1,2]. While inflammation is a critical part of an effective response to harmful stimuli, inappropriate acute or chronic inflammatory signaling can cause harm to the body. Widespread adoption of inflammation management protocols has helped to lower the death rates, but there are still many inflammatory disorders for which there are no specific approved treatments. As a result, new therapeutic approaches are being pursued. An emerging strategy involves microRNAs (miRNAs), which have been shown to play significant roles in inflammation in general and in specific inflammatory conditions such as sepsis, both in promoting pathogenesis as well as recovery [3,4,5,6,7,8,9,10,11,12,13,14,15]. For example, miRNAs such as miR-146a and miR-223 have been shown to be downregulated in both septic vs. healthy patients and in non-surviving vs. surviving patients [4]. Thus, the concept of therapeutic miRNA delivery is intriguing as a possible novel anti-inflammatory treatment.

When considering potential vehicles for miRNA delivery, extracellular vesicles (EVs) have been implicated as promising based on their reported natural ability to facilitate intercellular RNA transfer [16]. While the physiological significance of EV-mediated miRNA transfer is still controversial [17,18,19], the capabilities of specifically-loaded EVs for small RNA delivery (siRNA and miRNA) have been clearly established [20,21,22]. Further, direct comparisons of EVs and other potential miRNA delivery vehicles such as liposomes have indicated the potential superiority of EVs [23,24,25]. Thus, EV-mediated miRNA delivery to treat inflammation is worthy of focused investigation.

Here, we built on previous work from our group using a sonication-based miRNA loading strategy to package miRNA into EVs without requiring chemical modifications [26]. Our prior study, similar to many in the field to date, investigated delivery of only a single miRNA species. In this work, we sought to exploit the potential synergy of regulating multiple anti-inflammatory pathways by loading multiple miRNA species into a single EV population. Combinations of miRNAs were tested in an in vitro macrophage inflammation model, which was previously shown to correlate with in vivo outcomes for EVs [27]. Finally, the most effective combination was tested in an in vivo endotoxemia model.

## 2. Results

### 2.1. EV Loading and Characterization

To test multiple different combinations of miRNAs, a method of sonication-mediated EV loading previously developed by our lab was employed [26]. The sonication method is an exogenous loading technique in which pre-synthesized siRNA or miRNA mimics can be mixed in any combination to EVs and loaded, with minimal damage to both the EVs and RNA [26]. EVs derived from HEK293T cells were collected and the ability to controllably co-load two different miRNA cargos into a single EV population was determined by mixing and sonicating Cy3-labeled miR-93 and Cy5-labeled miR-126 in varying proportions (Figure 1A). These two RNAs, which were selected for this experiment due to the availability of tagged species in our laboratory that made detection more feasible, were found to be loaded near their input proportion as determined via fluorescence after washing away the excess RNA (Section 4.5). The sonicated EVs were characterized via western blot (Figure 1B; full blots available in Appendix A), nanoparticle tracking analysis (NTA) (Figure 1C), and transmission electron microscopy (TEM) (Figure 1D) according to the recommendations of the International Society for Extracellular Vesicles [28].

### 2.2. Screening for Anti-Inflammatory miRNA

As an initial assessment of anti-inflammatory bioactivity, the effect on IL-6 secretion was selected as a screening criterion based on a prior report that showed correlation between the effects of EVs on IL-6 secretion in vitro and their anti-inflammatory activity in vivo [27]. HEK293T EVs were chosen for study due to their expected limited anti-inflammatory bioactivity as well as low intrinsic RNA content [29]. EVs were loaded with three different miRNA mimics (miR-146a, miR-155, and miR-223) that were found in the literature to be downregulated in septic patients, to regulate the TLR4 inflammatory pathway, and/or to have altered expression levels in response to LPS stimulation [4,30,31,32,33,34,35,36,37,38]. While certain single-stranded miRNAs (including miR-146a-5p) have been shown to be proinflammatory TLR agonists [3,9,39,40], these double-stranded mimics are designed to interact with the RNA-induced silencing complex (RISC) with preferential strand selection. The miRNAs were loaded either individually, in combination with another, or as the complete group. In this way, each miRNA could be compared with others both as a mono-treatment and when left out of the complete group. These EV treatments were applied to RAW264.7 murine macrophage cells for 24 h, when the supernatant was replaced by LPS treatment for 4 h, in a “pre-treat” regime. At the end of the LPS treatment, the supernatants were collected, assessed using an IL-6 ELISA, and compared to the “No miRNA” group, the EVs sonicated without miRNA present. All treatment groups led to significant anti-inflammatory effects, with dose-dependence evident (Figure 2A). Interestingly, the No miRNA group (unmodified HEK293T EVs) showed an anti-inflammatory effect on par with 10 μg/mL dexamethasone (Dex), reflecting the prior data showing the benefits of HEK293T EVs in a sepsis model via an unknown mechanism [41]. Cell phagocytic behavior was tested after LPS treatment to see if EV-mediated miRNA treatment impaired phagocytosis (Figure 2B). No significant changes were detected, indicating that treatments were not inducing endotoxin tolerance. Due to the effectiveness of each miRNA combination, more challenging regimes were employed to differentiate between the combinations.

All the groups were next tested in a “co-treat” regime, wherein LPS and EV treatments were both applied concurrently to RAW264.7 cells for 24 h (Figure 3A). miR-146a alone had a significant anti-inflammatory effect, while miR-223 alone and miR-155 alone had no effect. In contrast, strikingly, the 155/223 combination significantly reduced the IL-6 secretion. The 146a/223 treatment was not significantly effective, while the 146a/155 and 146a/155/223 treatments significantly reduced the IL-6 secretion. Next, all groups were tested in a “post-treat” regime, wherein LPS was applied concurrently to RAW264.7 cells for 24 h, and then the LPS and EV treatments were concurrently applied for 24 h (Figure 3B). In this regime, no significant effects were detected except for with the complete combination of 146a/155/223. Finally, a murine endotoxemia model was employed to assess the 146a/155/223 combination. IL-6 serum levels in mice dosed with 146a/155/223 showed a 23% reduction in cytokine plasma concentration compared to the LPS control animals. (*p* = 0.07) (Figure 3C). Interestingly, a serum IL-6 reduction associated with the control group, HEK EVs loaded with cel-miR-67 Negative Control miRNA mimic (“NC”), was also observed, once again reflecting the previous data showing an unexpected anti-inflammatory benefit of HEK293T EVs [41].

### 2.3. Delivery of miRNAs 146a/155/223 Has Variable Anti-Inflammatory Effects Aside from Reducing IL-6 Secretion

Given the effectiveness of the 146a/155/223 combination in suppressing IL-6 secretion, we screened to see if other relevant secreted cytokines were also regulated using an antibody-based cytokine array. Pretreatment of RAW264.7 cells with EV-delivered 146a/155/223 or NC showed differential protein expression after LPS treatment for 4 h (Figure 4A; full data set available in Appendix A). In comparison to the NC, 146a/155/223 induced downregulation of IL-6, IL-10, CCL22, CCL17, CXCL10, CXCL13, and CXCL16 (Figure 4B). Array data for all targets are available in Appendix A. CCL22 downregulation in vitro was verified via ELISA (Figure 4C). However, 146a/155/223 treatment failed to induce any change in CCL22, TNFa, MIP-2, or IL-1β secretion in endotoxemic mice as compared to the NC or LPS-only controls (Figure 4D).

## 3. Discussion

We previously established that sonication enables the loading of miRNA into EVs with only slight diminishment of in vitro EV uptake compared to unmodified EVs [26]. In this study, we demonstrated that the loading of two different small RNA sequences by sonication was predictable based on the proportion of their concentration in solution, and that loading of three distinct miRNAs can potentially yield improved anti-inflammatory bioactivity compared to a single miRNA delivery. This technique may thus allow several advantages over competing EV loading strategies. Any mixture of miRNA sequences can potentially be loaded into a single EV population with a reproducible loading efficiency, though it is possible some sequences may behave in different fashion. As opposed to mixtures of singly loaded EVs, the premixing of miRNA allows for the possibility of loading multiple miRNAs into a single vesicle, promoting proportional delivery to a recipient cell. This exogenous loading technique is also adaptable for any small RNA cargo and does not require any manipulation of the cargo or producer cells. Future tests, including for the specific combination of miR-155, miR-223, and miR-146a, should be performed to determine the percentage of loaded EVs and if any specific subpopulation of EVs is preferentially loaded by this technique.

To take advantage of this system, we performed a screen for anti-inflammatory miRNA combinations using a limited number of miRNAs selected from the literature. These miRNA combinations were passed through progressively more rigorous LPS challenges in vitro to determine if any specific combination of miRNAs was superior in reducing inflammation. That process identified the combination of miR-146a, miR-155, and miR-223 as being the most efficacious amongst the examined groups in reducing IL-6 production by RAW264.7 macrophages in response to the LPS. This finding echoes work by Bhaskaran et al. that found that overexpression of three miRNAs in glioblastoma had a combinatorial anticancer effect [42], as well as a clinical study by Marik et al. which found that a combination of hydrocortisone, ascorbic acid, and thiamine worked synergistically as an anti-inflammatory against sepsis [43,44].

miR-146a, miR-155, and miR-223 have been studied as anti-inflammatory miRNAs that change expression levels in response to LPS and target proteins in the TLR4 pathway [30,31,32,45]. Interestingly, these miRNA targets are largely nonoverlapping, perhaps indicating that when attempting to downregulate a cellular pathway, greater effect may be achieved by targeting different proteins in that pathway rather than focusing on one protein. Work by Schulte et al. described the tiered response by macrophages to LPS, in which miR-146 expression saturated at even sub-inflammatory LPS concentrations in order to protect against hypersensitivity, whereas miR-155 was expressed proportionally over a broad range of LPS concentrations in order to respond appropriately to the level of stimulation [46]. This indicates that both miRNAs seem to work in tandem to prevent an extreme cellular response. However, in other contexts, introducing miR-155 has been shown to be proinflammatory [47,48]. For example, EVs from wildtype bone marrow-derived dendritic cells (BMDCs) increased IL-6 production in response to LPS in miR-155^−/−^ BMDCs and mice, compared to EVs from miR-155^−/−^ BMDCs [49]. These seemingly contradictory results indicate that miR-155 activity is nuanced and likely context dependent. Concurrent introduction of other anti-inflammatory miRNAs such as miR-146a and miR-223 may tilt the RNA network towards an environment in which miR-155 suppresses inflammation.

The results of our protein array showed a downregulation of IL-6, as expected. CCL22 and CCL17, the two CCR4 ligands, which are involved in T-cell chemotaxis, were also downregulated by the 146a/155/223 treatment. Interestingly, in an LPS challenge model, CCR4-deficient mice had decreased cytokine release and a higher survival rate when compared to wildtype mice [50]. In another study, CCR4-deficient mice had reduced immune response and greater survival after cecal ligation and puncture (CLP) and greater responsiveness and survival to a secondary fungal challenge [51]. These previous results indicated that, in addition to inhibition of IL-6, inhibition of the CCR4 ligands CCL22 and CCL17 may lead to an improved outcome in vivo.

Despite these encouraging signs, we saw no significant decrease in proinflammatory cytokines in response to 146a/155/223 in vivo. There are multiple reasons why this may be the case. Firstly, since cell source plays a role in EV biodistribution and delivery [52], the choice of HEK293-derived EVs may limit an in vivo effect. A recent study showed that HEK293 EVs have a short half-life in healthy mice; in one hour, 80% of EVs were cleared from the circulation [53]. It is possible that cargo packaged within EVs from mesenchymal stromal cells or another cell source could have a greater chance of functional delivery. Additionally, while sonication may inhibit EV delivery only slightly in vitro, this effect may be increased under more challenging delivery conditions in vivo. Finally, the in vitro model used to screen for anti-inflammatory effects may be insufficiently representative of in vivo dynamics, despite the prior correlation noted in the literature [27]. For example, the RAW264.7 macrophage model may be insufficiently representative of native macrophage behavior and is certainly insufficiently representative of other cell types affected by LPS injection.

## 4. Materials and Methods

### 4.1. Cell Culture

Human embryonic kidney HEK293T cells and RAW264.7 mouse macrophage cells (ATCC, Manassas, VA, USA) were cultured in Dulbecco’s modified Eagle’s medium (DMEM; R&D Systems, Minneapolis, MN, USA) supplemented with 10% EV-depleted fetal bovine serum (FBS; ThermoFisher, Waltham, MA, USA) and 1% penicillin/streptomycin (ThermoFisher, Waltham, MA, USA) in T175 tissue culture polystyrene flasks. The FBS was EV-depleted via 100,000× *g* centrifugation at 4 °C for 16 h, where the supernatant was retained.

### 4.2. Extracellular Vesicle Isolation

The conditioned media were collected and subjected to differential centrifugation. Briefly, the supernatant was centrifuged at 1000× *g* for 10 min, 2000× *g* for 20 min, and 10,000× *g* for 30 min, after each of which the supernatant was retained, and finally, it was centrifuged at 100,000× *g* for 2 h, after which the pellet was resuspended in PBS and collected. The final spin was performed using an Optima L-90K ultracentrifuge with T70i rotor (Beckman Coulter; Sykesville, MD, USA). This resuspension was washed 2× using Nanosep 300-kDa MWCO spin columns (Pall; Port Washington, NY, USA). The washed EVs were resuspended in PBS and filtered using an 0.2 μm syringe filter. The EV size distribution and concentration were determined by nanoparticle tracking analysis (NTA) via a NanoSight LM10 (Malvern Panalytical; Westborough, MA, USA). Each sample was analyzed in triplicate using consistent acquisition settings. The total EV protein was determined via bicinchoninic acid assay (BCA) following the manufacturer’s protocol. The relative levels of the relevant protein components were determined via western blotting. The samples were electrophoresed on a 4–15% polyacrylamide gel on a Mini-PROTEAN Tetra Cell (Bio-rad; Hercules, CA, USA) and transferred to nitrocellulose using the Trans-Blot transfer system (Bio-rad). Alix (Abcam (Shanghai, China): ab186429), TSG101 (Abcam; ab125011), GAPDH (Cell Signaling Technology, Danvers, MA, USA; 2118L), and CD63 (ThermoFisher; 25682-1-AP) primary antibodies were added at a 1:1000 dilution, except for GAPDH (1:2000). Secondary antibody IRDye 800CW anti-Rabbit (LI-COR Biosciences (Lincoln, NE, USA) (926-32211) was added at 1:10000 dilution, and the membranes were imaged on a LI-COR Odyssey CLX Imager.

### 4.3. Extracellular Vesicle Loading

First, 100 µg EVs, corresponding to ~3 × 10^9^ particles detected by NTA, were mixed with 1 nmol miRNA mimic, and the volume was brought up to 100 μL with PBS. In preparations with more than one miRNA mimic, mimics were added in equal proportion to reach 1 nmol. This mixture was incubated for 30 m at room temperature, before being sonicated in a water bath sonicator (VWR^®^ symphony™; 97043-964) (Swedesboro, NJ, USA) (2.8 L capacity, dimensions 24 L × 14 W × 10 D cm) at 35 kHz for 15 s, placed on ice for 1 m, and sonicated for a second 15 s. The mixture was placed back on ice briefly, then washed 3× using Nanosep 300-kDa MWCO spin columns to remove the unincorporated RNA, and resuspended by PBS. The miRNA mimics (Dharmacon; Lafayette, CO, USA) used were: hsa-miR146a-5p (C-300630-03); hsa-miR-155-5p (C-310430-07); hsa-miR-223-3p (C-300580-07); and Negative Control #1 (C-310391-05). For sonicated EVs without the miRNA mimic added, PBS was added instead of the RNA.

### 4.4. Transmission Electron Microscopy (TEM)

The EVs were negatively stained using a protocol, as previously described [54]. Briefly, 4% paraformaldehyde (10 μL) was added to the EVs (10 μL), which incubated for 30 min. A carbon film grid (Electron Microscopy Sciences; Baltimore, MD, USA; CF200-Cu-25) was placed on the paraformaldehyde/EV droplet for 20 min and washed with PBS. Then, the grid was placed on 1% glutaraldehyde (50 μL) for 5 min and washed eight times with water. Finally, the grid was placed on uranyl acetate replacement stain (50 μL) for 10 min and left to dry for 10 min. The images were acquired on a JEOL JEM 2100 LaB6 TEM at 200 kV (40,000× magnification) using a digital camera (Gatan; Pleasanton, CA, USA).

### 4.5. Fluorescent-Labeled RNA Co-Loading Test

Pre-labeled Cy3-labeled miR-93 (Dharmacon; CTM-433488) and Cy5-labeled miR-126 (Dharmacon; CTM-508110) were mixed at the indicated ratios and loaded according to the sonication protocol discussed in Section 2.3. After extensive washing, fluorescence readings were acquired, normalized using a standard curve, and compared as a fraction of the total fluorescence.

### 4.6. In Vitro RAW264.7 Inflammatory Assay

RAW264.7 cells were seeded in DMEM supplemented with 5% FBS in a 48-well plate at 100,000 cells per well. All EVs were prepared by sonication, and doses were normalized by protein content after sonication and washing. All treatments were diluted in DMEM supplemented with 5% FBS. In the “pre-treat” regime, cells were treated with EVs for 24 h; then, the supernatant was replaced by media with 10 ng/mL lipopolysaccharide (LPS) (Sigma-Aldrich, St. Louis, MO, USA; L4391) for 4 h. In the “co-treat” regime, both EV treatments and 10 ng/mL LPS were added concomitantly for 24 h. In the “post-treat” regime, cells were treated with 10 ng/mL LPS for 24 h and then 10 ng/mL LPS with EV treatments for another 24 h. As a negative control for each experiment, both no EVs (PBS only) and EVs sonicated without miRNA were added to cells. Dexamethasone (10 μg/mL) (Sigma-Aldrich; D4902) was added as a positive control. After all final treatments, the media were collected and stored at −80 °C. The IL-6 concentration was determined using the Mouse IL-6 DuoSet ELISA Kit (R&D Systems; DY406). For the “co-treat” regime, phagocytosis was measured after the removal of the media, using the Vybrant Phagocytosis Assay Kit (Invitrogen, Carlsbad, CA, USA; V-6694) and following the manufacturer’s protocol. Briefly, fluorescein-labeled *E. coli*-derived particles were added to cells for 2 h, after which cell fluorescence was measured in a plate reader. All tests were performed in biological triplicate.

### 4.7. ELISA

Cytokine concentrations were determined via DuoSet ELISA Kits (R&D Systems): IL-6 (DY406), TNFa (DY410), MIP-2 (DY452), IL-1β (DY401), and CCL22 (DY439). Plasma samples were diluted 100-fold for all cytokines, except for TNFa which was diluted tenfold.

### 4.8. Proteome Array

An antibody-based protein array was performed on the cell supernatants after a “pre-treat” regime, using the Proteome Profiler Mouse XL Cytokine Array (R&D Systems’ ARY028) according to the manufacturer’s protocol. The expression (pixel density) was normalized between membranes using positive and negative reference spots on each membrane.

### 4.9. In Vivo Endotoxemia Study

Male C57BL/6J mice (Jackson Labs, Bar Harbor, ME, USA), 8 to 12 weeks of age, were used in this study. The animals were kept at a constant temperature (25 °C) under a 12 h light/dark cycle with free access to food and water. On the first and second day, animals received a 200 μL intraperitoneal injection of PBS or sonicated EVs at a concentration of 2.1 × 10^10^ particles/mL (by NTA). On the third day, animals received an intraperitoneal injection of 5 mg/kg LPS. Three hours later, animals were anesthetized and sacrificed via cardiac blood collection. Blood was collected into EDTA-coated tubes (Greiner Bio-One; Monroe, NC, USA) and spun at 1000× *g* for 15 min to produce plasma. All animal work was carried out in accordance with the NIH guidelines and approved by the Institutional Animal Care and Use Committee (IACUC) at the University of Maryland College Park.

### 4.10. Statistical Analysis

Data are presented as mean ± SD. One-way ANOVAs with Dunnett’s multiple comparison test were used to determine the statistical significance in the in vitro inflammatory assay and the in vivo endotoxemia experiments. All statistical analysis was performed with Prism 8 (GraphPad Software, La Jolla, CA, USA).

## 5. Conclusions

Sonication is an effective method for loading multiple miRNAs into EVs in predictable proportions. Given the vast number of targets that are regulated by any one miRNA sequence, it would be difficult to fully map or predict the changes in the transcriptome, proteome, or phenotype of a cell that takes up one miRNA, let alone three. In this way, while the literature can guide the selection of therapeutic miRNA, empirical combinatorial testing of multiple miRNAs may be necessary when seeking to design an miRNA-based therapeutic. This work, which by no means exhausts the possible space of miRNA combinations, is nonetheless our attempt to illuminate the strengths of such an approach.

## Figures and Tables

**Figure 1 ncrna-08-00071-f001:**
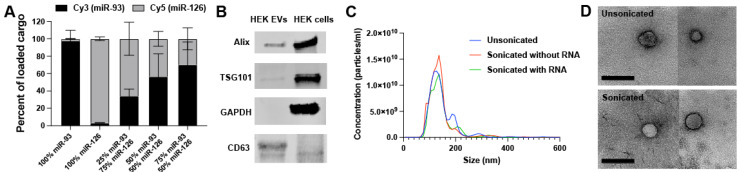
EV characterization and co-loading validation. (**A**) Relative quantification of co-loaded fluorescently tagged miRNA mimics. (**B**) Western blot of EVs vs. parental cells. (**C**) Nanoparticle Tracking Analysis (NTA) performed on EVs that were unsonicated, sonicated, and sonicated with miRNA. (**D**) Transmission electron micrographs (TEM) of unsonicated and sonicated EVs. Scale bar = 200 nm.

**Figure 2 ncrna-08-00071-f002:**
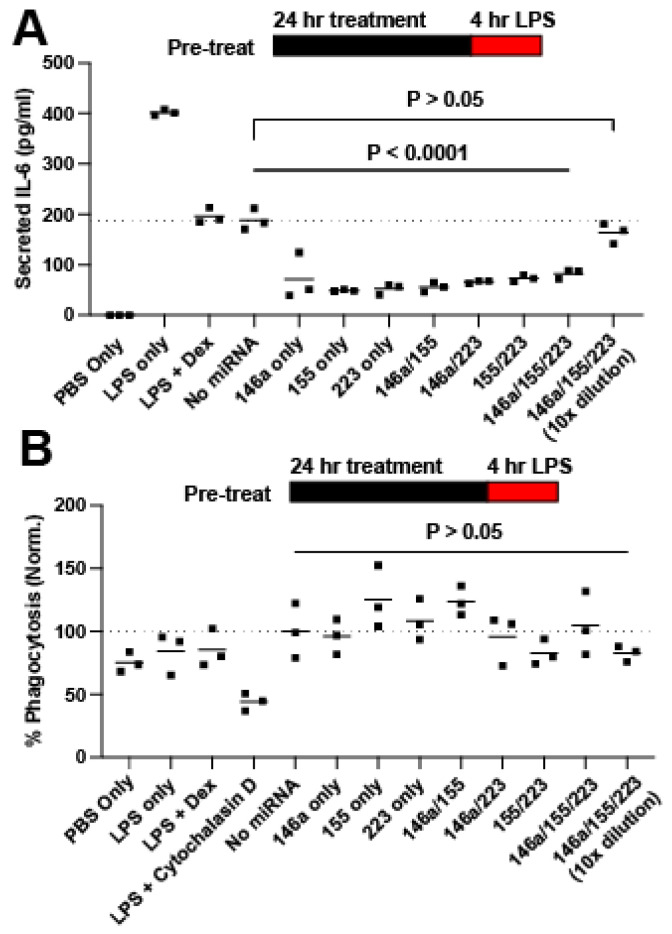
Screening of miRNA for anti-inflammatory combination. (**A**) Secreted IL-6 in response to LPS in a pretreatment regime. (**B**) Phagocytosis as measured by the Vybrant Phagocytosis Assay Kit (Invitrogen).

**Figure 3 ncrna-08-00071-f003:**
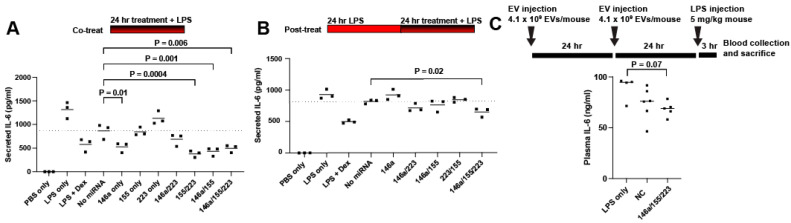
Screening of miRNA for anti-inflammatory combination. (**A**) Secreted IL-6 in response to LPS in a co-treatment regime. (**B**) Secreted IL-6 in response to LPS in a post-treatment regime. (**C**) IL-6 plasma levels in endotoxemic mice. Results were analyzed via one-way ANOVA.

**Figure 4 ncrna-08-00071-f004:**
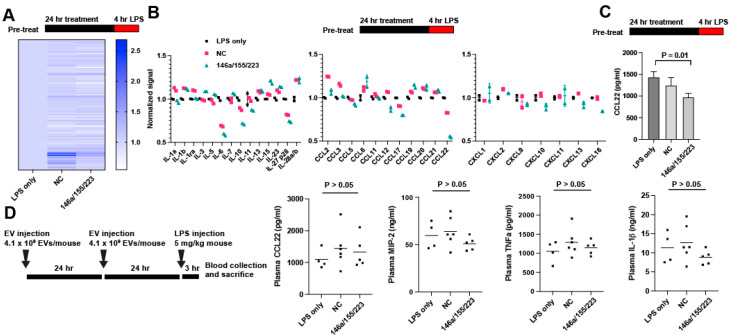
Screen for extracellular protein targets of the miR 146a/155/223 combination. (**A**) Relative expression for all targets as measured by the Proteome Profiler Mouse XL Cytokine Array (R&D Systems) (*n* = 2). (**B**) Relative expression for the IL, CCL, and CXCL cytokines. (**C**) CCL22 expression was quantified via ELISA. Results were analyzed via one-way ANOVA. (**D**) Treatment schedule and serum cytokine levels for endotoxemic mice pretreated with 146a/155/223 or NC as indicated. Results were analyzed via one-way ANOVA.

## Data Availability

The normalized protein array data displayed in Figure 4A are available in Appendix A. Any other data is available from the authors upon request.

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
