# Peer review of "Combinatorial microRNA Loading into Extracellular Vesicles for Increased Anti-Inflammatory Efficacy"

_ncrna, 2022, doi:10.3390/ncrna8050071_

Round 1

Reviewer 1 Report

This manuscript aims to demonstrate that loading EVs with a combination of miRNAs via sonication is an effective technique for miRNA delivery and for potential use as EV-based therapeutics using inflammation as an example. While this is an exciting and very relevant topic, the overall manuscript leaves this reviewer with more questions than insights as it covers a lot of ground from EV characterization following miRNA loading, to in vivo cytokine production, without covering any of the topics in depth. This reviewer appreciates the inclusion of EV characterization as recommended by ISEV but would like to see additional data regarding EV miRNA loading if that is the focus of the manuscript. This reviewer in not convinced that figure 1 demonstrates that miRNA loading is predictable and results in efficient EV loading of multiple EVs.  

Additional thoughts are listed below:

·         While figure 1 shows the relative quantification of EVs loaded with two RNA mimics, the remaining figures use EVs loaded with up to three mimics at a time. Data should be presented to include the quantification of EVs when loaded with three mimics. Do the approximate percentages change? And is loading sequence specific? Figure 1 provides data for unrelated miRNAs while the reminder of the publication focuses on miRs-146a, 155, and 223.

·         Bulk analysis methods for EV characterization can be a bit misleading as EVs are highly heterogenous and difficult to separate. It would be helpful to understand what percentage of the total EV population contain multiple EV mimics as cargo. Characterization at the single EV level would help address this.   

·         The n in Figure 4A and B should be clarified.  

·         The methods section needs additional details to enhance rigor and reproducibility. This includes (but is not limited to) additional information regarding TEM data acquisition procedures, fluorescent labeling of RNA (particularly with respect to normalization and determination of percentages), Dex information, ELISA dilutions, proteome array normalization, and most importantly, negative controls.

·         Formatting could also be improved. Examples include lines 70-71, and the citation at line 94.

Reviewer 2 Report

Dear authors, it was a pleasure to read your work. It is executed at a high methodological level and carefully written. It needs only minor additions and explanations, mainly in the Materials and Methods section.

Section 2.3 should clearly state the proportion of RNAs used in final preparations. Has this ratio been confirmed by the method described in Section 2.5? It is also required to expand section 2.3 with a clear description of the production of unsonicated vesicles, sonicated RNA-free vesicles, and sonicated RNA-containing vesicles. Clearly indicate which manipulations were performed on which vesicles.

Section 2.4 does not specify the characteristics of the microscope with which the studies were carried out. And the characteristics of a digital camera. I also wanted to clarify whether the work was carried out on carbon films or on formvar ones. Such a thorough fixation requires an explanation.

Section 2.5 should be expanded with the vesicles used, loading method, RNA ratio and reagents used. Or included in section 2.3.

Section 2.6 requires you to expand the abbreviation LPS the first time you use it. Explain, please, the sentence on line 114. At what point does transfection occur? It may be worth moving transfection into a separate section.

Line 71 has an extra line break character. On line 274 it is unacceptable to break «-\-», the appropriate character must be used - non-breaking space.

The abbreviation «NoTx» used in Figure 2 should be deciphered in the text or in the caption.

Of interest is the question of what RNAs are contained in extracellular vesicles of HEK cells and in what concentration? Are the authors sure that the concentration of RNA they used significantly exceeds that in cell vesicles in order to discuss the effects of artificially introduced RNA?

Reviewer 3 Report

Comment 1: In the article entitled “Combinatorial microRNA loading into extracellular vesicles for increased anti-inflammatory efficacy”, an established approach to extrinsic nucleic acid loading of EVs, sonication, was utilized to enable controlled loading of multiple miRNAs in HEK293T EVs. Combinations of carefully chosen miRNAs were compared to single miRNAs with respect to anti-inflammatory outcomes in assays of increasing stringency, with the combination of miR-146a, miR-155, and miR-223 found to have the most potential amongst tested groups.

Presented data in this paper are important for inflammation and EVs focused groups. Although the authors give much evidence for their conclusions, the experimental design and the choice of control group is not reasonable, and additional experiments are required to increase the robustness of the conclusions.

That is why I can recommend this paper for publication only after a major revision of below comments.

Comment 2: Introduction: Page 1 line 37-39: It would be useful to introduce more articles addressing the reasons why these three miRNAs were chosen, especially miRNA-155. Afterall, based on recent findings, miR-155 shows a pro-inflammatory effect.  

Comment 3: Materials and Methods: Page 3 line 101: Why miR-93 and 126 were used in this part of the experiment, but not the three miRNAs in the manuscript?

Comment 4: Materials and Methods: Page 3 line 114: “Cell transfection was achieved” --- Has the cell transfection technique been used in this manuscript? I also, do not see any description of the phagocytosis method in this manuscript.

Comment 5: Materials and Methods: Please be consistent and add or remove catalogue numbers for all compounds/media/reagents mentioned in the manuscript. Please use the same format for all reagents.

Comment 6: Results: Figure1/A: “Relative quantification of co-loaded fluorescently-tagged miRNA mimics” --- Please specify which method was applied in Fig1A and add it to the Materials and Methods.

Comment 7: Results: The miRNA/miRNAs expression level in EVs loaded with miR-146a, miR-155, miR-223 or a combination should be measured by qPCR. Meanwhile, preferably compared to the NC group rather than the No miRNA group.

Comment 8: Results: Fig.1/B – There is representative pictures of western blot of proteins from cell and EV. However, nothing is mentioned in Methods how western blot was performed. Please add this paragraph to the Materials and Methods.

Comment 9: Results: Figure2/A: Please define “Dex” and “Tx”. Moreover,

Please describe how the EVs in the no-miRNA group were treated and whether they were ultrasonicated. And why did not use the NC group as a control.

Comment 10: Results: Please standardize the terms in the manuscript. (For example, the combination group is used with three different terms in the manuscript, All, 146a/155/223 and TRI, respectively.)

Comment 11: Results: Figure3/C: In this figure, there is no statistical difference between groups ctrl and 146a/155/223. It is not appropriate to use a comparison between LPS only and 146a/155/223 groups. After all, EV itself has anti-inflammation effect according to your data shows in the manuscript.

Comment 12: Results: Figure4/D: Please delete “p=0.07”.

Round 2

Reviewer 1 Report

Thank you for the thoughtful responses. The ultimate goal of this publication is two parts, to demonstrate that sonication is an effective technique that can be utilized to load EVs with a combination of miRNAs in a predictable manner (which is the bulk of the focus). And secondly, that these EVs can be used for therapeutic drug delivery. However, EV loading and characterization is only represented in 1/4 of the figures while it is the focus of the manuscript title, abstract and much of the text. Although microRNA mimics are relatively the same size and structure, and the results would likely be as expected, I would argue that a clear characterization of EVs after loading three miRNA mimics would be necessary in order to support your statement on line 255 that "In this study, we showed that the loading of multiple small RNA sequences by sonication is predictable based on the proportion of their concentration in solution." 

Author Response

Thanks for the clarification. We have now changed the opening paragraph of the discussion section to more precisely convey what our results show.

"We previously established that sonication enables the loading of miRNA into EVs with only slight diminishment of in vitro EV uptake compared to unmodified EVs [27]. In this study, we demonstrated that the loading of two different small RNA sequences by sonication is predictable based on the proportion of their concentration in solution, and that loading of three distinct miRNAs can potentially yield improved anti-inflammatory bioactivity compared to single miRNA delivery. This technique may thus allow several advantages over competing EV loading strategies. Any mixture of miRNA sequences can potentially be loaded into a single EV population with a reproducible loading efficiency, though it is possible some sequences may behave in different fashion. As opposed to mixtures of singly-loaded EVs, the pre-mixing of miRNA allows for the possibility of loading multiple miRNAs into a single vesicle, promoting proportional delivery to a recipient cell. This exogenous loading technique is also adaptable for any small RNA cargo and does not require any manipulation of the cargo or producer cells. Further tests could determine the percentage of loaded EVs and if any specific subpopulation of EVs is preferentially loaded by this technique."

Reviewer 3 Report

Comment 1

 Please continue to revise according to the Comment 12, I understand that “p=0.07” is present in 3C but should not be present in 4D.

Author Response

Apologies, I did not notice this even the second time. It has now been removed from 4D.

Round 3

Reviewer 1 Report

Lines 21 and 315 should also be re-worded. 

Author Response

Lines 21 and 315 should also be re-worded. 

Line 21 now reads: Here, an established approach to extrinsic nucleic acid loading of EVs, sonication, was utilized to load multiple miRNAs in HEK293T EVs. 

Line 315 now reads: That process identified the combination of miR-146a, miR-155, and miR-223 as being the most efficacious amongst the examined groups in reducing IL-6 production by RAW264.7 macrophages in response to LPS.